# Metacognitive Precursors: An Analysis in Children with Different Disabilities

**DOI:** 10.3390/brainsci7100136

**Published:** 2017-10-21

**Authors:** María Consuelo Sáiz Manzanares, Miguel Ángel Carbonero Martín

**Affiliations:** 1Department of Health Science, University of Burgos, 09001 Burgos, Castilla y León, Spain; 2Department of Psychology, University of Valladolid, 47002 Valladolid, Castilla y León, Spain; miguelangel.carbonero@uva.es

**Keywords:** metacognition, precursors, validation scale, disabilities

## Abstract

The analysis of Metacognitive skills is a key element to guide the learning process. Current research has shown the initiation of these skills from an early age. The present study had two aims: (1) to validate a Scale Measuring Precursor Metacognitive Skills (SMPMS) in children with diverse disabilities, and (2) to study possible significant different between different disabilities in precursor metacognitive skill use. We worked with 87 children with different disabilities, with an average age range of 24–37 months. The results have shown high indicators of reliability and validity of the SMPMS. We isolated two factors related to cognitive and metacognitive and self-regulation skills response to an adult. We also found significant differences in the acquisition of metacognitive and self-regulation skills among children with global developmental retardation as compared to children with expressive language and comprehension disability.

## 1. Introduction

Metacognition is a concept that has been introduced by Flavell [1], and it means to engage in reflection about the own cognition. Also, Brown [2] introduced the relation between metacognition with self-regulated concept. Second, it provides the tools to learn metacognitive skills around human development. In the human learning, first the self-regulated skills will be introduced to adult, and later these will be internalized by the subject. Various models in the scientific literature have attempted to explain the development of metacognitive skills, among them, the sequential model of Zimmerman [3]. This model focuses on the relationship between planning strategies prior to task execution, orientation strategies during task execution, and the assessment strategies after task completion, also include a social perspective and a regulation component [3,4]. Another relevant model is that of Nelson and Narens [5], based on task-solving processes and on feedback about the resolution circuit. This model examines the influence of the metamemory processes that are employed during task resolution and the relationship between the levels of metacognitive and cognitive access—object-level, referring to the current cognitive task being solved, and meta-level, which includes the mental representation of the object-level. The meta-level represents monitoring (i.e., the object-level information about the meta-level) and control when information descends from this level to the object-level. Strategy correction of erroneous execution takes place at the meta-level.

In young children, the study of initiation in metacognitive skills has focused on the analysis of individual skills. Metacognitive precursors (according to Cambridge Dictionary “precursor” is “something that happened or existed before another thing, especially if it either developed into it or had an influence on it”. Specifically, in this paper precursor concept will apply from J.C Gómez [6] research about cognitive and metacognitive skills. They have the they are related to the capacity to represent and symbolize.) are related to the development of the skills regarding social comprehension, emotional comprehension, comprehensive and expressive language, and development of symbolic play [7,8]. Recent research [9] has found a relationship between symbolic play development and behavioral self-regulation processes [7,8,10]. In these processes, situational comprehension and the development of planning are critical for task-solving strategies [11]. The acquisition of these skills helps children to acquire self-knowledge [12]. These strategies are essential for task resolution, especially those that involve putting oneself in the other’s place [13,14,15], although the development of these skills can be altered by developmental problems. There also appears to be relationship between cognitive development and planning and prediction skills [7,15]. These skills are metacognitive precursors and predictors of effective learning [12]. Likewise, the development of self-knowledge seems to be related to the development of the adult skills of planning and response to adult regulation and to the subsequent acquisition of one’s own self-regulation in problem solving [16]. All of these skills can be learned and are therefore susceptible to training at early ages [17,18,19,20].

Another relevant issue is to specify when the acquisition and differentiation of these strategies begins [20]. There has been traditionally a dichotomous position: considering that cognition directs metacognition [21] or, conversely, when considering that metacognition directs cognition [19]. Recent research [22] is oriented towards a joint understanding of that relationship. Hence, cognitive development may be a determinant of the acquisition of some of the metacognitive skills, although these skills do not seem to follow a uniform evolutionary pattern of acquisition [7,23]. In fact, the investigations of the development of mentalist skills (directly related to the metacognitive skills of planning and assessment) in subjects with disabilities carried out by Rivière, García-Nogales and Nuñez [24] pointed out that a deficit in the acquisition of these skills was also detected in generalized developmental disorders such as maturation delay, language delay, especially of comprehensive language, and cognitive retardation [7]. Also, recent research has found differences in the self-regulation behaviors of children with various disabilities [25].

Another handicap encountered is the evaluation of metacognitive skills. It is currently considered that they can be measured through two methods [26,27]: online methods (in situ analysis is performed while the subject performs the task) and offline methods (analyses are carried out after the subject has reflected on the performed tasks). Offline methods require the development of the skills of self-awareness, comprehensive and expressive language, and long-term memory processes [15,27,28,29,30]. When considering these aspects, the most appropriate procedures for the analysis of metacognitive skills and their precursors in children are the online methods during task resolution [8,31].

The evaluation of metacognitive skills in young children is a challenge for cognitive and developmental psychology. These disciplines have encountered difficulties to measure them. Quantitative and qualitative evidence [12,32] has been methodologically limited, as subjects’ responses must be collected individually and in detail. These data collection techniques make it difficult to work with large samples [8,12,27,33]. Therefore, one of the main problems is to calculate the validity and reliability indices of the measuring instruments due to the reduced *n* [12,34,35].

In the other hand, there are other theoretical paradigms that dispute the existence of metacognitive precursors without the development of language, such as Perner’s [36]. He considers that the skills described above are not properly metacognitive skills, he calls it MiniMetaCognition.

Taking the above into account, the objectives of this study were:Research Issue 1: To determine the reliability and validity indices of the “Scale Measuring Precursor Metacognitive Skills (SMPMS)” in children with disabilities.Research Issue 2: To determine the functional relationship between metacognitive precursors in children and different types of disability.

## 2. Materials and Methods

### 2.1. Participants

The following development quotients, using the Brunet-Lezine Revised Test of Psychomotor Development in Early Childhood-see Instruments [37] were obtained for all of the participants: Psychomotor Development Quotient (PDQ), Cognitive Development Quotient (CDQ), Expressive Language Development Quotient (ELDQ), Comprehensive Language Development Quotient (CLDQ), and Socialization Development Quotient (SDQ). We worked with 87 subjects: 36 boys and 51 girls (see Table 1).

Group 1. Global developmental delay (GDD) with GDQ between 55 and 70, as measured with the Brunet-Lezine Revised Test of Psychomotor Development in Early Childhood (BLRT). This group had 29 subjects: 12 boys and 17 girls.Group 2. GDD with GDQ between 70 and 80, measured with the BLRT. This group included 16 subjects: 7 boys and 9 girls.Group 3. Comprehensive Language Disorder (CLD) with GDQ between 55 and 70, as measured with the Reynell Developmental Language Scales III (RDLS-III) [38]. This group comprised 15 subjects: 6 boys and 9 girls.Group 4. Psychomotor Delay (PD) with GDQ between 60 and 80, as measured with the BLRT. This group was made up of 12 subjects: 4 boys and 8 girls.Group 5. Expressive language disorder (ELD) with GDQ between 55 and 70, as measured with the RLDS-III. This group included 15 subjects: 7 boys and 8 girls.

These children were being treated at three early-care centers of a Castile and Leon city. Sample selection was not random, but instead, we used incidental sampling. The allocation of the children to the categories was carried out according to the criteria of the Diagnostic and statistical manual of mental disorders-fifth edition (DSM-5) [39], classification for Groups 1 and 2, and of the *International* Classification of Diseases (ICD-10) [40], for Groups 3, 4, and 5, because the DSM-5 does not differentiate between expressive and comprehensive language. The evaluation was carried out after obtaining the family’s authorization to participate in the early intervention program. The families had medium socio-economic status.

### 2.2. Research Design

A descriptive-correlational design was used to test the hypothesis in the Research Issue 1. To verify the hypothesis in the Research Issue 2, we used a pre-experimental design without a control group, in which the independent variable was the type of disorder and the dependent variable was the metacognitive precursors [41].

### 2.3. Instruments

#### 2.3.1. Brunet-Lezine Revised Test of Psychomotor Development in Early Childhood (BLRT)

This scale analyzes the level of development by areas: Psychomotor area (test-retest reliability = 0.92), Cognitive area (test-retest reliability = 0.75), Communication-Language (test-retest reliability = 0.82), and Autonomy-Socialization (test-retest reliability = 0.50); global test-retest reliability = 0.89. The test also establishes development quotients in the different areas.

#### 2.3.2. Reynell Developmental Language Scales III (RDLS-III)

These scales examine the development of Comprehensive and Expressive Language and establish the corresponding development quotients (comprehensive language: Kuder-Richarson reliability = 0.97, and expressive language: Kuder-Richarson reliability = 0.96).

#### 2.3.3. Symbolic Play Test (SPT)

This test analyses the development of symbolic play from 12 to 36 months of age. It consists of 24 items of observation of play situations. The score is dichotomous (1 if the action is performed and 0 if it is not), with a maximum of 24 points. The raw scores are equivalent in developmental ages, ranging from less than 12 months to more than 36 months.

#### 2.3.4. Scale Measuring Precursor Metacognitive Skills (SMPMS)

This is a probabilistic scale of (Likert-type) summary estimates, in which the evaluators must choose from five options the frequency with which the behavior was observed. In its original version, it had 24 items but after validation, it was reduced to eight items explained by two factors (see Appendix A
Table 7).

### 2.4. Procedure

We assessed each child in three 45-min individual sessions. In the first session, we applied the BLRT [37] and the SMPMS (The SMPMS it is an observational scale and it was applied when children solving the task of the BLRT. We considered adult self-regulation when the evaluator had the sentences for child was the tasks of BLRT). In the second session, we applied the RDLS-III [38]. Finally, the SPT [42] was applied in the third session. After obtaining authorization from the families, the sessions were recording in order to subsequently study the participants’ responses in detail. The tests were administered by a specialist in early childhood assessment who was an extern psychologist to the early care centers. The tests were applied within the Diversity Attention Program -DAP- of the Meeting of Castile and Leon, it is a Program to assessment different handicaps in children throughout their schooling in early age (0 to 72 months). The informed consent of all families and directors of a center were collected in writing.

### 2.5. Data Analysis

Given that the participants were chosen as a function of availability in the early care centers (0–3 years), we performed an a priori analysis of multivariate normality on the obtained BLRT scores [33] with the Bollen-Stine [43] bootstrap procedure, accepting the hypothesis of normality (*p* = 0.11). To verify the first hypothesis, we conducted an Exploratory Factor Analysis (EFA), using the principal axes factoring method and oblimin rotation, the composite reliability test, and the Average Variance Extracted (AVE). Subsequently, we performed a Confirmatory Factor Analysis (CFA) using the Maximum Likelihood Method (MLM). For the second hypothesis, we performed a fixed effects single-factor (type of disorder) Analysis of variance (ANOVA) and Tukey’s posttest. The analyses were conducted with Statistical Package for the Social Science-version 24 (SPSS v.24) and AMOS v.23.

## 3. Results

### 3.1. Study 1

Regarding Research Issue 1 (determine the reliability and validity indices of the SMPMS in children with disabilities), content validity of the SMPMS was estimated through expert opinion (10 experts, psychologists specialized in children’s early assessment [0–6 years], with more than 10 years of experience, and aged between 40 and 55 years). They analyzed the (semantic) content and form (syntax) of the items of the scale, rating from 1 (Not at all) to 5 (Completely). In accordance with the results of the analysis, four items were eliminated: Item 4.1. (The child reflects on the process of task resolution), Item 4.2 (The child can reflect on the response to a task and, if it is wrong, corrects it with a slight intervention by the adult), Item 4.3 (The child can reflect on the response to a task and, if it is wrong, corrects it spontaneously), and Item 4.4 (The child knows what kind of strategies to use to solve a task) because these behaviors would be difficult to analyze due to the developmental characteristics of the target population. We then performed Exploratory Factorial Analysis (AFE) to determine whether more than one factor could be extracted. Subsequently, to verify the homogeneity of the scale, we calculated the correlations between the items. We used a Pearson correlation matrix, as recent studies [43] indicate that it is the most appropriate when the sample contains less than 200 individuals. We found significant correlations between several items and no significant correlations between some of them. These results showed that more than one factor could be extracted. The correlation coefficients ranged between *r* = 0.20 and *r* = 0.96, except for Items 2.5 (The child can emit elements of a phrase or short phrases, but without discourse), 3.1 (The child employs expressive language consistent with the proposed task), 3.3 (The child performs autonomous actions of lengthy process without prior planning), and 3.5 (The child performs complex activities of lengthy process with a perfect plan). If items had correlations of *r* = 0.90 or higher, we proceeded to select one of them as a function of the global nature of the wording to avoid multicollinearity (see Table 2). The SMPMS was thus reduced to nine items. We conducted EFA on these items, obtaining the following indicators: Kaiser-Meyer-Olkin (KMO) = 0.74, Bartlett’s sphericity test = 1372.79, *p* = 0.05, cumulative variance extracted = 68.57%, and global reliability index α = 0.94. Using the principal axes factorization method with oblimin rotation, we extracted two factors that explained 62.70% of the variance. Table 3 shows the relationship between the factors and the corresponding factor loadings. Through the EFA (see Table 3), we extracted two factors, Factor 1 comprises the metacognitive precursors, and Factor 2 includes language (comprehensive and expressive) response to regulation, and removed Item 2.5 (“The child can emit elements of one phrase or short phrases, but without discourse”) because its factor loading was lower than 0.43, so the scale was finally reduced to eight items. The two factors explained 74.97% of the variance (the first factor explained 62.12% and the second factor 12.84%).

To determine the fit of the model, we conducted CFA. Previously, we studied the skewness and kurtosis of the distribution. Skewness values higher than |2.00| indicate extreme asymmetry, and lower values indicate that the distribution is normal [44]. Kurtosis values between |8.00| and |20.00| indicate extreme kurtosis [45,46]. We found skewness values ranging between |44| and |1.17|, and kurtosis values between |0.02| and |1.96|, suggesting that there is no severe deviation from normality in any of the items (see Table 4).

All of the standardized regression coefficients (factor loadings) between the items and the extracted dimensions in metacognitive strategies had high positive values, between 0 and 1 [47], indicating a link between the factors and items associated with them. This implies a robust factorial structure. To determine whether the assumed model presented a good fit, we used the ML estimation method, which assumes multivariate normality of the data. The fit indices used were: chi-square test (*χ^2^*), CMIN = minimum discrepancy divided by degrees of freedom (*df*), normed fit index (NFI), comparative fit index (CFI), Tucker-Lewis index (TLI), root mean square error of approximation (RMSEA), Akaike Information criterion (AIC), and parsimonious fit index (ECVI) [46,48,49]. The empirical verification of the fitted models can be seen in Table 5. These values confirm the two factors extracted from the EFA. Figure 1 shows the confirmatory factorial relationship and loadings of each of the scale items on the corresponding factor. For the first factor, we found AVE = 0.67, and composite reliability (CR) of 0.90, and for the second factor, AVE = 0.66, and CR = 0.67. For the latent variables, CR, which indicates the internal consistency of the construct, was calculated with the following formula [50] (p. 130).

This index should be calculated for each construct, and the obtained value should be equal to or higher than 0.70. The higher the construct reliability, the greater the internal consistency of its indicators. Therefore, the indicator of CR for the first factor was very adequate, and that of the second factor was within acceptable margins. We calculated AVE by applying the following formula [50]. As the indicators become more representative of the latent construct, the value of the variance increases, and the recommended value is ≥0.50. Hence, the AVE indicators were within the recommended range.

In both of the formulas *λ^2^_j_* represents the standardized coefficient of each indicator with the construct, and *ξ_j_* the error of measurement associated with indicators.

It can therefore be concluded that the SMPMS presents psychometrically acceptable indicators of reliability and validity.

### 3.2. Study 2

The aim was to determine the functional relationship between metacognitive precursors in children and the different types of disability. As can be seen in Table 6, we found significant differences in the comprehension of externally motivated phrases, that is, regulated by the adult (Item 2.2), in the emission of phrases (Item 2.6), and in the performance of behaviors following the adult’s regulation (Item 3.2), with an AVE in of *η^2^* = 11, *η^2^*= 0.22, and *η^2^* = 0.20, respectively.

We performed Tukey’s post-hoc test to determine between which groups differences were found, detecting significant differences the following items: Item 2.2 between Groups CLD and ELD in favor of the latter (mean difference = −0.90, *p* = 0.02); Item 2.6 between Groups GDD (55–70) and GDD (70–80) in favor of the latter (mean difference = −1.26, *p* = 0.31), between Groups GDD (55–70) and PD (70–80) in favor of the latter (mean difference = −1.26, *p* = 0.01), and between Groups GDD (55–70) and ELD in favor of the latter (mean difference = −0.87, *p* = 0.02); and, in Item 3.2 between Groups GDD (70–80) and ELD in favor of the latter (mean difference = −1.06, *p* = 0.001).

Summarizing, we found significant differences in the use of metacognitive skills depending on the kind of impairment. Children with less cognitive impairment used them more frequently when solving tasks.

## 4. Discussion

Knowledge of metacognitive precursors is relevant in psychological and educational contexts, as these skills are related to effective learning. However, the scales or inventories that can measure them tend to have low reliability and validity indices. Especially at early ages, online assessment methods should be used [12,34] as they allow for observation of children’s behaviors during task resolution, although this also makes it more difficult to work with large numbers of subjects [12]. This aspect is more pronounced when children have some disability [7,33,35]. This study provides a scale (SMPMS) in which we found psychometrically acceptable indicators of reliability and validity, although the results must be taken with caution because the number of participants was not very high and they were not selected randomly. This supports the hypothesis that metacognitive precursors are related to the development of the skills of comprehensive and expressive language [7,8]. Hence, this instrument will enable professionals in the fields of psychology, pedagogy, and occupational therapy to analyze metacognitive precursors at early ages, and, depending on the results, develop individualized training programs [17,18].

This work also provides empirical evidence of the functional relationship between the skills in response to adult regulation and language comprehension skills. Subjects with no cognitive or language comprehension problems responded more to adult behavioral regulation [7,9]. This can be explained by the problems of individuals who have comprehensive and cognitive impairments when they use attentional and planning skills to solve problems [7,11,20]. The results indicate that adult regulation, whether behavioral or verbal, seems to be related to cognitive development, language comprehension skills, and metacognitive precursors. The response to verbal regulation has been found to be better in children who only have a deficit in the development of expressive language and in children with less cognitive impairment. The situational comprehension of a task is directly related to children’s level of cognitive development and skills in response to adult regulation and it facilitates their development of planning strategies and self-knowledge [7,10,11,20]. These skills are predictors of successful learning [12] and are linked to the subsequent acquisition of self-regulation in learning processes [16]. These skills can be learned, and are therefore susceptible to training at early ages [17,19,20]. Hence, the detection of deficits is important to perform adapted training programs [17,18].

Also, these findings support the mixed interpretation of the relationship between cognition and metacognition and the differentiation of metacognition in self-regulation skills [19,22,25].

## 5. Conclusions

The behavior regulation it is very important to cognitive and metacognitive acquisition at an early age. Also, the level of cognitive and language competencies is a good predictor to learning strategies. Likewise, the limitations of this study are related to the number of participants, 87. All of the studies on the acquisition and development of metacognitive skills in young children have this difficulty [8,12,20] because it is difficult to carry out microanalytical measurements with many participants. However, future research will be aimed at increasing the current number both in populations with impairments and in “standard” populations.

The analysis of the results will allow us to determine the kind of behaviors and responses to adult regulation that are developed in children with different pathologies. This aspect is particularly relevant in the fields of education and therapeutic intervention. Therefore, further studies will be aimed at the creation of intervention programs for the development of metacognitive skills at early ages adjusted to the identified difficulties and adapted to the needs of different disabilities [17,18]. In these researches, we will be use the control group to analyze whether the early self-regulated programs improve metacognitive skills. Also, in further research, we will analyze in longitudinal studies that there is relationship between dates of precursor Metacognitive skills (according to Perner [36] MiniMetaCognition skills) and metacognition skills e.g., Theory of Mind skills.

## Figures and Tables

**Figure 1 brainsci-07-00136-f001:**
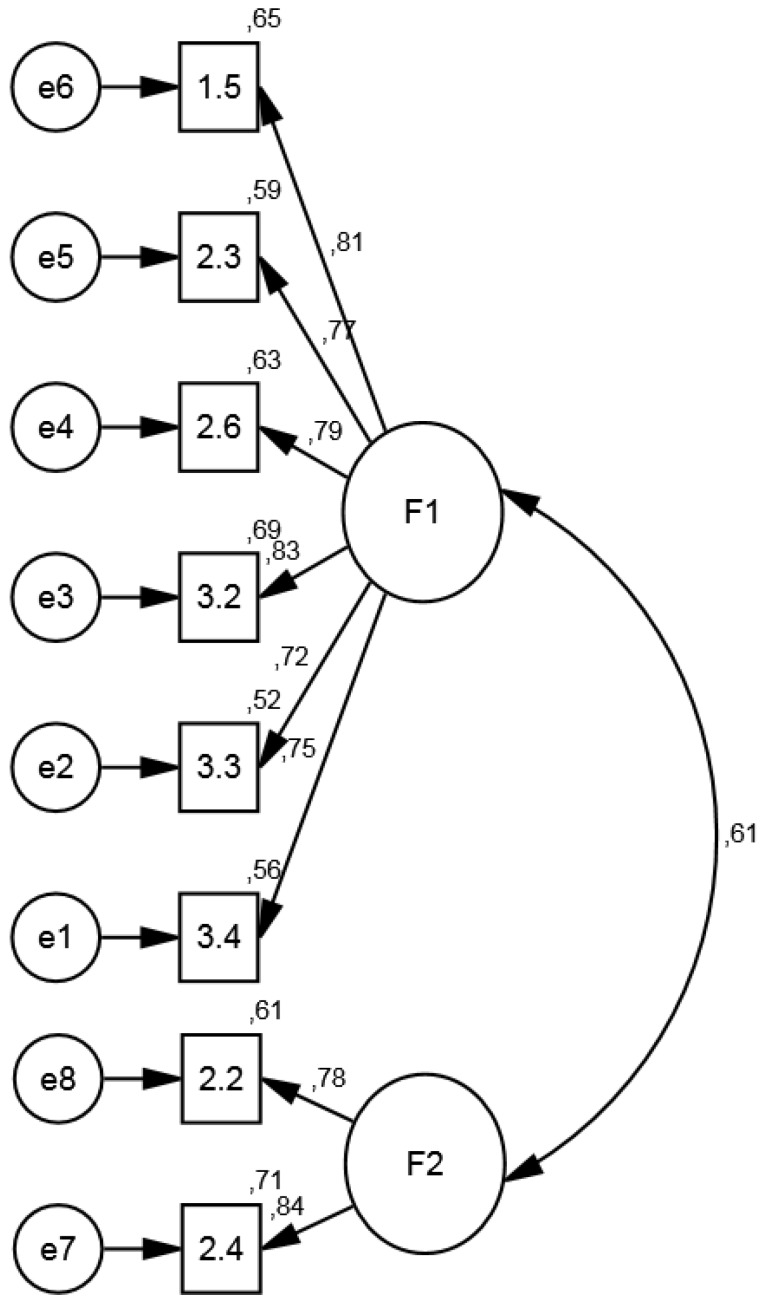
Multiple squared correlations and standardized regression weights. F_1_ = Factor 1; F_2_ = Factor 2.

**Table 1 brainsci-07-00136-t001:** Participants´ Mean and Standard Deviation (SD) in Brunet-Lezine and Reynell Quotients.

Descriptive Statistics and Ranges	Boys	Girls
Mean Age	30.40 months	30.95 months
SD Age	7.02	4.55
Age range	25–77 months	25–77 months
Mean _PDQ_	70.17	67.12
Mean _CDQ_	63.91	64.28
Mean _ELDQ_	59.57	66.36
Mean _CLDQ_	62.81	66.86
Mean _SDQ_	65.63	66.71
Quotient range	55–77	55–75

Note. The participants were grouped according to various categories (handicaps). PDQ = Psychomotor Development Quotient; CDQ = Cognitive Development Quotient; ELDQ = Expressive Language Development Quotient; CLDQ = Comprehensive Language Development Quotient; SDQ = Socialization Development Quotient.

**Table 2 brainsci-07-00136-t002:** Inter-element Correlation Matrix and Means and Standard Deviations of the **s**cale of assessment of metacognitive precursors (SMPMS).

Items	1	2	3	4	5	6	7	8	9	10	11	12	13	14	15	16
1.1 The child maintains sustained attention when the adult emits a message.																
1.2 The child maintains sustained attention when the adult proposes a task.	0.96 **															
1.3. The child maintains sustained attention when a task is performed by the adult.	0.92 **	0.93 **														
1.4 The child maintains sustained attention when spontaneously starting to perform a task.	0.91 **	0.94 **	0.93 **													
1.5 The child maintains sustained attention when the adult proposes a task.	0.88 **	0.88 **	0.88 **	0.90 **												
2.1. The child understands simple sentences of three elements.	0.80 **	0.79 **	0.78 **	0.82 **	0.69 **											
2.2 The child understands sentences of six elements and carries out the externally motivated actions.	0.38 **	0.37 **	0.42 **	0.44 **	0.35 **	0.44 **										
2.3 The child understands a discourse.	0.64 **	0.61 **	0.66 **	0.57 **	0.58 **	0.59 **	0.20									
2.4 The child can issue short sentences but without discourse while performing externally motivated actions.	0.44 **	0.43 **	0.46 **	0.48 **	0.53 **	0.37 **	0.65 **	0.22								
2.5 The child can emit elements of one phrase or short phrases, but without discourse.	0.32 *	0.28 *	0.34 **	0.32 *	0.33 *	0.29 *	0.37 **	0.52 **	0.06							
2.6 The child can emit simple sentences with limitations in the topic of conversation.	0.68 **	0.65 **	0.67 **	0.69 **	0.66 **	0.61 **	0.26	0.61 **	0.32 *	0.44 **						
3.1. The child employs expressive language consistent with the proposed task.	−0.003	0.08	−0.02	0.09	−0.09	0.19	0.30 *	−0.15	0.19	−0.25	−0.30 *					
3.2. The adult’s language can direct short functional actions.	0.71 **	0.71 **	0.69 **	0.74 **	0.63 **	0.80 **	0.41 **	0.63 **	0.50 **	0.15	0.65 **	0.15				
3.3 The child performs autonomous actions of lengthy process without prior planning.	0.57 **	0.55 **	0.54 **	0.61 **	0.57 **	0.57 **	0.48 **	0.57 **	0.49 **	0.60 **	0.51 **	−0.08	0.55 **			
3.4. The child performs externally motivated actions without prior planning.	0.54 **	0.50 **	0.53 **	0.55 **	0.59 **	0.50 **	0.28 *	0.59 **	0.20	0.51 **	0.60 **	−0.24	0.65 **	0.54 **		
3.5. The child performs complex activities of lengthy process, with a perfect plan.	0.37 **	0.33 *	0.37 **	0.38 **	0.46 **	0.26	−0.04	0.44 **	−0.03	0.46 **	0.40 **	−0.16	0.36 **	0.44 **	0.66 **	
Mean (M)	2.46	2.45	2.32	2.34	2.27	2.45	2.05	2.09	1.98	1.82	2.05	2.16	2.23	1.96	1.84	1.77
Standard Desviation (SD)	0.93	0.89	0.93	0.94	0.92	1.14	0.75	0.96	0.86	0.69	0.92	1.00	0.95	0.80	0.73	0.79

* *p* < 0.05. ** *p* < 0.01.

**Table 3 brainsci-07-00136-t003:** Factors and Factor Loadings of the SMPMS Items in each Factor.

Items	F_1_	F_2_
1.5 The child maintains sustained attention when the adult proposes a task.	0.82	0.07
2.2 The child understands sentences of six elements and carries out the externally motivated actions.	0.43	0.64
2.3 The child understands a discourse.	0.63	0.39
2.4 The child can issue short sentences but without discourse while performing externally motivated actions.	0.46	0.70
2.5 The child can emit elements of one phrase or short phrases, but without discourse.	0.40	−0.15
2.6 The child can emit simple sentences with limitations in the topic of conversation.	0.79	−0.11
3.2. The adult’s language can direct short functional actions.	0.79	0.19
3.3 The child performs autonomous actions of lengthy process without prior planning.	0.80	−0.22
3.4 The child performs externally motivated actions without prior planning.	0.59	−0.43

Note. F_1_: comprises the metacognitive precursors; F_2_: includes language (comprehensive and expressive) response to self-regulation.

**Table 4 brainsci-07-00136-t004:** Analysis of the Normality of the SMPMS.

Items	Minimum	Maximum	Skewness	CI	Kurtosis	CI
1.5 The child maintains sustained attention when the adult proposes a task.	1	5	0.56	1.77	0.16	0.24
2.2 The child understands sentences of six elements and carries out the externally motivated actions	1	4	0.44	1.34	0.08	0.12
2.3 The child understands a discourse.	1	5	1.07	3.27	1.34	2.05
2.4 The child can issue short sentences but without discourse while performing externally motivated actions.	1	4	0.72	2.20	0.02	0.04
2.6 The child can emit simple sentences with limitations in the topic of conversation.	1	5	1.01	3.10	1.00	1.53
3.2 The adult’s language can direct short functional actions	1	5	1.17	3.59	1.24	1.89
3.3 The child performs autonomous actions of lengthy process without prior planning.	1	4	0.90	2.74	0.79	1.21
3.4 The child performs externally motivated actions without prior planning.	1	4	1.09	3.34	1.96	3.00

Note. CI = critical interval.

**Table 5 brainsci-07-00136-t005:** Goodness-of-Fit Indices.

		Two-Factor Model (Pre-Determined Model)	One-Factor Model	Accepted Value
	*df*	20	19	
	χ^2^/*df*	28.59	-	
Residual-based indices	CMIN/*df*	1.50	7.18	
	RMSEA	0.06	0.21	(0.05, 0.08)
	RMSEA confidence interval	(0.00, 0.10)	(0.19, 0.24)	
	SRMR	0.06	-	0.05–0.08
Comparative fit index	NFI	0.90	0.00	0.90–0.95
Proportion of variance indices	CFI	0.96	0.00	0.95–0.97
	TLI	0.91	0.00	0.85–0.90
Indices of degree of parsimony	AIC	78.59	274.56	The lowest value
	ECVI	0.59	2.06	
	ECVI interval (90%)	0.52–0.73	1.70–2.47	

Note. CMIN = minimum discrepancy divided by *df*; NFI = normed fit index; CFI = comparative fit index; TLI = Tucker-Lewis index; RMSEA = root mean square error of approximation; AIC = Akaike Information criterion; ECVI = parsimonious fit index.

**Table 6 brainsci-07-00136-t006:** Descriptive Statistics and Analysis of variance of one factor fixed effects (different types of disability) (ANOVA) and Value of the Effect on Students with Different Pathologies in the SMPMS.

Items	Group 1 GDD 55–70	Group 2 GDD 70–80	Group 3 CLD	Group 4 PD	Group 5 ELD	*F*(4, 82)	*p*	*η^2^*
	*n* = 29	*n* = 16	*n* = 15	*n* = 12	*n* = 15			
	M (SD)	M (SD)	M (SD)	M (SD)	M (SD)			
1.5. The child maintains sustained attention when the adult proposes a task.	2.10 (0.77)	2.75 (1.00)	2.33 (0.97)	2.81 (0.87)	2.70 (1.54)	2.50	0.05	0.11
2.2. The child understands sentences of six elements and carries out the externally motivated actions.	2.06 (1.03)	2.06 (0.68)	1.73 (0.45)	2.00 (0.77)	2.64 (0.49)	2.60	0.04 *	0.11
2.3. The child understands a discourse.	1.86 (0.78)	2.37 (0.88)	2.40 (1.35)	2.81 (0.87)	2.42 (1.08)	2.30	0.07	0.10
2.4. The child can issue short sentences but without discourse while performing externally motivated actions.	2.06 (1.03)	2.18 (1.04)	2.06 (0.80)	1.45 (0.52)	2.35 (0.92)	1.60	1.83	0.07
2.6. The child can emit simple sentences with limitations in the topic of conversation.	1.55 (0.46)	2.56 (1.09)	2.20 (0.94)	2.00 (0.00)	3.00 (1.10)	5.70	0.00 **	0.22
3.2. The adult’s language can direct short functional actions.	1.93 (0.59)	1.75 (0.57)	3.33 (0.57)	2.20 (1.09)	2.83 (1.16)	4.99	0.01 **	0.20
3.3. The child performs autonomous actions of lengthy process without prior planning.	2.00 (0.75)	2.12 (0.81)	1.93 (0.96)	2.54 (0.93)	2.57 (1.01)	1.81	0.13	0.08
3.4. The child performs externally motivated actions without prior planning.	1.68 (0.47)	2.18 (0.91)	1.80 (0.77)	2.00 (0.00)	2.07 (0.82)	1.82	0.13	0.08

Note: GDD = Global Developmental Delay; CLD = Comprehensive Language Disorder; PD = Psychomotor Delay; ELD = Expressive Language Disorder; M = Mean; SD = Standard Deviation; *p* = probability; *η^2^* = eta square it is effects value. **p* ≤ 0.05. ** *p* ≤ 0.01.

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
