# Peer review of "Metacognitive Precursors: An Analysis in Children with Different Disabilities"

_brainsci, 2017, doi:10.3390/brainsci7100136_

Round 1
Reviewer 1 Report
Manuscript number/title: brainsci-223948 Metacognitive precursors: an analysis in children
with different disabilities
Journal: Brain Sciences
General comments
This manuscript, entitled “Metacognitive precursors: an analysis in children with different disabilities” focuses on an important area of development: early metacognition and precursor skills. Recently, this area of study has been gaining interest with both research and practitioners so this manuscript is timely. Moreover, the manuscript focuses on differences in metacognitive skills across young children with different disabilities which has important implications for early childhood education (practitioners and policy) as well as developmental science. The other aim of this manuscript is on validating an assessment instrument for measuring early metacognitive skills. This dual focus has the potential to make a strong contribution to the field integrating the conceptualization and assessment of early metacognitive and precursor skills and individual differences.
However, there are some areas of concern (outlined below) that need to be addressed before it can determined whether this manuscript warrants publication in Brain Sciences
Overall, the manuscript is in need of clarification and explication throughout both in terms of language / terminology usage and description / explanation including conceptualizations / operationalizations, the literature review, methodology, and procedures.
Specific comments:
Abstract:
In all, the abstract aligns with the manuscript and describes the study adequately. However, it would benefit from more precision of language and explication of what is meant by “precursors”, the study aims and methodology (just a brief sentence or two will do). Lastly, the first sentence was confusing- maybe “skills” needs to be possessive as in “Metacognitive skills’ analysis is..”. It should be made clear whether this sentence (and the authors’ focus is on measurement / assessment or conceptual / construct of metacognition and its precursors or both).
Highlights
Overall, the highlights were clear and described the study well. However, the last two need to be clarified: “Metacognitive precursors are slowed down in different disabilities” and “Metacognitive precursors are susceptible to training in early attention programs”. This also points to a larger issue that the “early attention program” was not described in the manuscript and it is not clear what this is and how it relates to the current study- was it an intervention given to the study children? Please clarify both in the Highlights and the body of the manuscript.
Introduction
My main concern about the introduction section is that there is a good amount of terminology that is not thoroughly defined / conceptualized. Metacognition is a construct that has been defined in many ways and thus it is crucial that the authors are explicit about the way that they are conceptualizing and operationalizing it. I have the same concern / comment for the authors’ use of precursor skills. If the authors are referring to something similar to Perner’s “Minimeta”, this should be addressed (either way, this concept should be discussed as it is extremely relevant to this paper). Further, there is no reference to Flavell, the originator of the term metacognition, which is concerning when the main focus of the manuscript is on measuring metacognition and comparing it across populations.
In all, the introduction needs to have a broader and deeper literature review as well as more explicit defining of the main study terms / concepts.
Method
My comments are similar for the method section as for the introduction. The methodologies employed are in need of explication and further description. It is unclear, for example, what the children were doing when the scales (i.e., BLRT; SMPMS; RDLS-III; SPT) were applied. Were they doing a problem-solving task? Were they in free play? Was this standardized across groups? And so on. It is also unclear when and how the adults interacted with the children
The authors were explicit about not having a control group but not about the reason why- this should also be included as it is an important limitation.
Not only is this explication important in terms of understanding the methodology and potential replication, but also for interpreting the results and their validity as well as implications.
Results
The result section is thorough and all tests conducted appear to be valid and explained in full.
However, it is not fully possible for me to be certain until the methods/procedures are comprehensively explained.
Discussion
The discussion was short and it was difficult to tell if it aligned with the introduction, methods, and results due to the issues I raised earlier- the limited clarity. The same issue exists in this section which needs to go into more depth regarding how the authors interpret their results and what contribution the results / this study makes to the field / implications. It would be more beneficial if this section was written more conceptually (rather than, for example, focused on the factor structure p. 10, line 264).
Also, on p. 10, line 273, please explicate / be more precise about what’s meant by “better” particularly in a metacognitive context. The “adult behavioral regulation” should also be explained (e.g., what did the adults do specifically- describe this in detain in the procedure section of the method section and then refer back to it here in the discussion section).
Conclusions, limitations and future lines of research:
On p. 10, line 289 this is an awkward sentence. This may be related to the limited precision / explication. Please explain further what is meant and clarify the language in this sentence too: “The behavior regulation it is very important to cognitive and metacognitive acquisition in early
age”.
Lastly, there was a discussion of limitations (such as the sample size) but no mention of not having a control group which is the greatest limitation in my view. This should be discussed and explained and, if the authors have a justification / reason they believe it did not affect the validity or interpretation of the results, this should be discussed as well.
Tables:
There were a good number of tables and some seemed to replicate information in text. However, I will leave this to the editor based on the journal specifications.
Author Response
Review 1
We thank the reviewer for his comments, which will certainly help us to improve our work.
Introduction
In all, the abstract aligns with the manuscript and describes the study adequately. However, it would benefit from more precision of language and explication of what is meant by “precursors”, the study aims and methodology (just a brief sentence or two will do). Lastly, the first sentence was confusing- maybe “skills” needs to be possessive as in “Metacognitive skills’ analysis is..”. It should be made clear whether this sentence (and the authors’ focus is on measurement / assessment or conceptual / construct of metacognition and its precursors or both).
Changes
Abstract: The analysis of Metacognitive skills analysis is a key element to guide the learning process. Current research has shown the initiation of these skills from an early age. The present study had two aims: 1) to validate a Scale Measuring Precursor Metacognitive Skills (SMPMS) in children with diverse disabilities and 2) to study possible significant different between different disabilities in precursor metacognitive skill use. We worked with 87 children with different disabilities, average age range of 24-37 months. The results have shown high indicators of reliability and validity of the SMPMS. We isolated two factors related to cognitive and metacognitive and self-regulation skills response to an adult. We also found significant differences in the acquisition of metacognitive and self-regulation skills among children with global developmental retardation compared to children with expressive language and comprehension disability
Highlights
Overall, the highlights were clear and described the study well. However, the last two need to be clarified: “Metacognitive precursors are slowed down in different disabilities” and “Metacognitive precursors are susceptible to training in early attention programs”. This also points to a larger issue that the “early attention program” was not described in the manuscript and it is not clear what this is and how it relates to the current study- was it an intervention given to the study children? Please clarify both in the Highlights and the body of the manuscript.
Change
We have eliminated the highlight “Metacognitive precursors are susceptible to training in early attention programs.”, because the review has raison in this manuscript we have not addressed the early stimulation programs directly. The only thing we wanted to point out is that through them you can improve these skills.
We explained that is DAP “is a Program to assessment different handicaps in children throughout their schooling in early age (0 to 72 months).”
Introduction
My main concern about the introduction section is that there is a good amount of terminology that is not thoroughly defined / conceptualized. Metacognition is a construct that has been defined in many ways and thus it is crucial that the authors are explicit about the way that they are conceptualizing and operationalizing it. I have the same concern / comment for the authors’ use of precursor skills. If the authors are referring to something similar to Perner’s “Minimeta”, this should be addressed (either way, this concept should be discussed as it is extremely relevant to this paper). Further, there is no reference to Flavell, the originator of the term metacognition, which is concerning when the main focus of the manuscript is on measuring metacognition and comparing it across populations. In all, the introduction needs to have a broader and deeper literature review as well as more explicit defining of the main study terms / concepts.
Changes
1. Metacognition is a concept that has been introduced by Flavell [1] and it means to reflection about the own cognition. Also, Brown [2] introduced the relation between this with self-regulated concept. Second it the tools to learn metacognitive skills around human development. First self-regulated skills were induced to adult and later these will be internalized by the subject.
2. food note and Definition of precursor according to Cambridge Dictionary “something that happened or existed before another thing, especially if it either developed into it or had an influence on it”. In this paper precursor concept will apply by J.C Gómez [6] research about cognitive and metacognitive skills.
Method
My comments are similar for the method section as for the introduction. The methodologies employed are in need of explication and further description. It is unclear, for example, what the children were doing when the scales (i.e., BLRT; SMPMS; RDLS-III; SPT) were applied. Were they doing a problem-solving task? Were they in free play? Was this standardized across groups? And so on. It is also unclear when and how the adults interacted with the children The authors were explicit about not having a control group but not about the reason why- this should also be included as it is an important
Changes
We introduced explanation about the SMPMS application.
Food note 2: The SMPMS it is an observational scale and it was applied when children solving the task of the BLRT.
At the end in the conclusions we inserted the sentence “In these researches, we will be use the control group to analyze if the early self-regulated programs improve metacognitive skills.”
limitation
Not only is this explication important in terms of understanding the methodology and potential replication, but also for interpreting the results and their validity as well as implications.
answer
The aims1 in this study was Research Issue 1: To determine the reliability and validity indices of the "Scale Measuring Precursor Metacognitive Skills (SMPMS)" in children with disabilities.
For this raison, we not applied control group.
The aim 2, was
“Research Issue 2: To determine the functional relationship between metacognitive precursors in children and different types of disability.”
Because we used pre-experimental design without control group, we have not applied never program, only we wanted to study to application of the scale.
Results
The result section is thorough and all tests conducted appear to be valid and explained in full.
However, it is not fully possible for me to be certain until the methods/procedures are comprehensively explained.
Changes
We explain more procedure sections.
Discussion
The discussion was short and it was difficult to tell if it aligned with the introduction, methods, and results due to the issues I raised earlier- the limited clarity. The same issue exists in this section which needs to go into more depth regarding how the authors interpret their results and what contribution the results / this study makes to the field / implications. It would be more beneficial if this section was written more conceptually (rather than, for example, focused on the factor structure p. 10, line 264). Also, on p. 10, line 273, please explicate / be more precise about what’s meant by “better” particularly in a metacognitive context. The “adult behavioral regulation” should also be explained (e.g., what did the adults do specifically- describe this in detain in the procedure section of the method section and then refer back to it here in the discussion section).
Change
Paragraph in p. 10 line 264 for “Also, we found that there were two factor, comprehension and expression language, that there were relations between adult verbal regulation” Paragraph in p. 10 273 we change “better” to “more”
We explained adult self-regulated in the second note food.
We considered adult self-regulation when the evaluator had the sentences for child was the tasks of BLRT.
Conclusions, limitations and future lines of research:
On p. 10, line 289 this is an awkward sentence. This may be related to the limited precision / explication. Please explain further what is meant and clarify the language in this sentence too: “The behavior regulation it is very important to cognitive and metacognitive acquisition in early age”. Lastly, there was a discussion of limitations (such as the sample size) but no mention of not having a control group which is the greatest limitation in my view. This should be discussed and explained and, if the authors have a justification / reason they believe it did not affect the validity or interpretation of the results, this should be discussed as well.
Changes
We have removed the sentence “The behavior regulation it is very important to cognitive and metacognitive acquisition in early age.”
We have not used the control group for reason explain before.
Tables:
There were a good number of tables and some seemed to replicate information in text. However, I will leave this to the editor based on the journal specifications.

Reviewer 2 Report
It details the validation of a scale to measure precursor metacogntive skills in children with different disabilities.Strengths of the study include the background and methodology used being clearly described, including the limitation of an small and incidental sample.
Weaknesses of the study include the restriction of parental regulation strategies to verbal ones. This would understandably have been expected to limit the ability of children with developmental disabilities to utilise these strategies, particularly if they had cognitive and language problems. No comment is made regarding the use of non-verbal parental regulation strategies; this would have been useful to place the findings in a more developmentally appropriate context. Although extensive statistical work has been undertaken to prove the validity and reliability of the scale, the conclusions can only really be considered preliminary given the nature of the sample. As a non-statistical expert, I would recommend that the manuscript is reviewed by someone with greater experience in this area.
Overall, I think this study would be of interest to readers interested in this field of research. Future research on this topic may also lead it to be of significance to clinicians working with infants and children with developmental disabilities.
Author Response
Review 2
Weaknesses of the study include the restriction of parental regulation strategies to verbal ones. This would understandably have been expected to limit the ability of children with developmental disabilities to utilise these strategies, particularly if they had cognitive and language problems. No comment is made regarding the use of non-verbal parental regulation strategies; this would have been useful to place the findings in a more developmentally appropriate context. Although extensive statistical work has been undertaken to prove the validity and reliability of the scale, the conclusions can only really be considered preliminary given the nature of the sample. As a non-statistical expert, I would recommend that the manuscript is reviewed by someone with greater experience in this area. Overall, I think this study would be of interest to readers interested in this field of research. Future research on this topic may also lead it to be of significance to
clinicians working with infants and children with developmental disabilities.
We thank the reviewer for his comments, which will certainly help us to improve our work. In futures researches we have an observation about the children responses to parents self-regulated behavioural.

Round 2
Reviewer 1 Report
I found this revised version of Metacognitive precursors: an analysis in children with different disabilities to be much improved and appreciate the clear indication of changes the authors made in each section.
However, I have a few remaining (minor) concerns that I believe can be addressed relatively easily and quickly. They are as follows:
Introduction
Though the authors have defined precursor, that does not address my concern. I believe that most readers will know the traditional definition. My concern was focused on how this term was being applied to metacognitive skills. The authors did include a reference (J.C Gómez). However, the important part, in my opinion, is what aspect of this research / paper is relevant. I don't believe that it's enough to write "In this paper precursor concept will apply by J.C Gómez [6] research about cognitive and metacognitive skills." In other words, it should not be a footnote as this is one of the main foci of the paper. This should, instead, be discussed in more detail- how Gómez used precursor (as a developmental antecedent or is it intended to mean that these skills are precursory to later skills typically referred to as metacognitive skills. In other words, do the authors conceptualize these precursor skills as "minimeta" type skills- Perner referred to such skills as not quite being able to be referred as metacognition proper. Or do they mean that these skills predict metacognitive skills?). Please clarify this point in the abstract (briefly), more thoroughly in the introduction, and come back to this in the discussion.
Discussion
The revised discussion was more aligned with the introduction, methods, and results. Though I still believe that this section should go into more depth regarding how the authors interpret their results and what contribution the results / this study makes to the field / implications. In other words, why is it important to study precursor skills in this way (vs. Perner's minimeta or other studies that focus on skills that predict metacognitive skills such as expressive language). This does not need to be extensive- a few key clarifying sentences will do.
Author Response
Dear Reviewer,
We send new article version. We hope that all the aspects you have indicated will be collected. We take this opportunity to thank you for your appreciations, which will undoubtedly improve the manuscript. The contributions of the second revision its are shown in yellow.
Best regards, and we remain at your disposal for any new modification.
María Consuelo Sáiz Manzanares
